# Metabolic impact of the VDR rs1544410 in diabetic retinopathy

**Caroline Severo de Assis**[1], **Tainá Gomes Diniz**[1], **João Otávio Scarano Alcântara**[1], **Vanessa Polyana Alves de Sousa Brito**[1], **Rayner Anderson Ferreira do Nascimento**[2], **Mayara Karla dos Santos Nunes**[3], **Alexandre Sérgio Silva**[4], **Isabella Wanderley de Queiroga Evangelista**[5], **Marina Gonçalves Monteiro Viturino**[5], **Rafaela Lira Formiga Cavalcanti de Lima**[6], **Darlene Camati Persuhn**[7] *

**1** Post-Graduate Program in Nutrition Science, Federal University of Paraiba, Joao Pessoa, Brazil, **2** Uninassau Faculty, João Pessoa, Brazil, **3** Post-Graduation Program in Development and Technological Innovation of Medicines (DITM), Federal University of Paraiba, Joao Pessoa, Brazil, **4** Department of Physical Education, Federal University of Paraiba, Joao Pessoa, Brazil, **5** Ophthalmology, Otolaryngology and Oral and Maxillofacial Surgery Unit, Lauro Wanderley University Hospital, Federal University of Paraiba, Joao Pessoa, Brazil, **6** Department of Nutrition and Post-Graduation Program in Nutrition Science, Federal University of Paraiba, Joao Pessoa, Brazil, **7** Department of Molecular Biology and Post-Graduation Program in Nutrition Science, Federal University of Paraiba, Joao Pessoa, Brazil

* darlenecp@hotmail.com

## Abstract

### Aims

To investigate the association between BsmI and DM2 in patients with and without DR and to correlate with clinical parameters in a population in northeastern Brazil.

### Methods

Cross-sectional case-control study in which data were collected from 285 individuals, including 128 patients with DM2 and 157 with DR. Clinical, biochemical and anthropometric parameters were analyzed, in addition to the single nucleotide polymorphism (SNP) BsmI of the VDR gene (rs1544410), genotyped by PCR-RFLP.

### Results

In the DR group we found a greater number of patients using insulin therapy (p = 0.000) and with longer duration of DM2 (p = 0.000), in addition to higher serum creatinine values (p = 0.001). Higher fasting glucose levels and higher frequency of insulinoterapy were independently observed in patients with DR and b allele carriers, when compared to BB.

### Conclusion

The association of the bb/Bb genotypes (rs1544410) of the VDR gene with increased blood glucose levels and insulinoterapy may represent worse glicemic control in rs1544410 b allele carriers in DR Latin American individuals.

**Data Availability Statement:** All relevant data are within the paper and its Supporting information files.

**Funding:** This study was supported by Public Call n. 005/2020 Programa Pesquisa para o SUS -

PPSUS - Paraíba State Research Foundation (FAPESQ, Paraíba, Brazil), National Council for Scientific and Technological Development (CNPq, Brasilia, Brazil) Ministry of Health / Decit/SCTIE (Decit/SCTIE/MS), State Health Secretary (SES/ Paraíba/Brazil); Grant 05/2021, Paraíba State Research Foundation (FAPESQ, Paraíba, Brazil). It was also supported by Public Call n. 03/2020 Produtividade em Pesquisa PROPESQ/PRPG/UFPB grant n. PIA13262-2020 and Coordination for the Improvement of Higher Education Personnel (CAPES) - Financial Code 001. The funders had no role in study design, data collection and analysis, decision to publish, or preparation of the manuscript.

**Competing interests:** The authors have declared that no competing interests exist.

## 1. Introduction

Diabetes mellitus is global epidemic health issue that affects around 415 million adults [1] Diabetic retinopathy (DR) is a frequent complication in type 2 Diabetes Mellitus (DM2) [2, 3], and clinical and metabolic factors are associated with the development and progression of DR [4–6].

There is evidence that deficiency of Vitamin D (VD) is related to DM2 [7, 8]. Vitamin D may have a direct effect on the function of pancreatic beta cells, mediated by the vitamin D receptor (VDR) [9], in addition, altered *VDR* gene transcription may influence fasting glucose levels by two potentially additive effects of vitamin D on adipocytes and pancreas cells [10, 11]. There is evidence that Vitamin D deficiency affects the pathogenesis and progression of DR and that patients with proliferative diabetic retinopathy (PDR) have lower levels of 25 (OH)D than those without diabetes [8, 12]. A meta-analysis encompassing 17,000 patients from several continents concluded that vitamin D deficiency increased the risk of DR in T2DM [13].

The relationship between vitamin D levels and DM2 and DR raises interest in investigating the effect of genetic aspects linked to this vitamin regarding its involvement in the etiology or modulation of DR, including single nucleotide polymorphisms (SNPs) in the Vitamin D receptor (VDR). The B allele of the BsmI polymorphism (rs1544410) of the *VDR* gene was associated with a lower risk of DR in Korean patients with DM2 [14], and the bb genotype was associated with a decrease in (25 [OH] D) in micro and macrovascular complications of DM2 in an Indian population [15]. However, there is no consensus in the literature on these relationships [13, 16–18], and the results seem to be influenced by the to geographic origin of patients.

Given the facts presented in the absence of data in the Brazilian population, we propose here to investigate the association between BsmI and DM2 in patients with and without DR and to correlate with clinical parameters using a case-control approach, in a population of northeastern Brazil.

## 2. Methods

### 2.1 Study design and ethical aspects

This cross-sectional case-control study was conducted in 285 patients recruited at the University Hospital Lauro Wanderley and the Basic Health Units of João Pessoa/Paraíba/Brazil. Data collection took place from 2014 to 2021 and patients invited to participate in the study signed an informed consent form. The project was approved by the Ethics Committee for Research with Human Beings of the Federal University of Paraíba (UFPB; Opinions No.: 796.459 (26/ 08/2014) and 3053068 (03/12/2018) ards of the institution and Resolution 466/2012 of the National Health Council.

Inclusion criteria were diagnosis of DM2 for at least 5 years, being in outpatient care. Exclusion criteria: diagnosis of DM1, insufficient DNA sample or with an inconclusive result in the genotypic analysis.

**2.1.1 Clinical characterization.** The diagnosis of DR was based on ophthalmoscopy after pupil dilation with 0.5% tropicamide. Images of the retina (macula and central disc) were captured at a 45˚ angle by a background camera. The images were analyzed according to the standards and recommendations of ACCORD (Action to Control Cardiovascular Risk in Diabetes) and the Early Treatment Diabetic Retinopathy Study (ETDRS). DM2 patients without DR (n = 128) constituted the control group and the group with some degree of retinopathy constituted the DR (n = 157). In the RD group, 49 patients had been diagnosed with DM2 for

up to ten years; the other 108 patients had been diagnosed with DM2 for more than ten years. Regarding the classification according to DR stage, 103 participants had non-proliferative diabetic retinopathy (NPDR) and 54 of them had proliferative diabetic retinopathy (PDR). Patients with a medical diagnosis of hypertension or who reported presenting this condition and were taking long-term medication for blood pressure control were considered hypertensive. The classification was performed according to the American Heart Association Guideline, American College of Cardiology, and the American Society of Hypertension (2015) [18].

## 2.2. Blood sampling

The blood samples were collected by venipuncture after night fasting. For biochemical analysis in general, blood in the presence of clot activator, for the determination of HbA1C in the presence of anticoagulant K3EDTA and for the determination of glucose in the presence of anticoagulant sodium fluoride, the samples were centrifuged at a speed of 3000 rpm for 10 minutes at room temperature for the separation of serum or plasma and subjected to analysis within 2 hours after collection, except for the sample of HbA1C that was analyzed in hemolyzed whole blood.

For DNA extraction, blood collection was performed by a venous puncture in sterile tubes containing 7.2 mg of K3 EDTA. Blood samples were stored for up to 20 days at -20˚C until DNA extraction was performed.

## 2.3. Biochemical analysis and anthropometric measurements

Enzymatic methods were used for total cholesterol, high-density lipoprotein (HDL) and triglyceride analyses. All the analyses were performed using an automated analyzer (Lab-Max 240, Labtest, Lagoa Santa, MG, Brazil) and standardized kits according to the manufacturer's instructions (Labtest, Lagoa Santa, MG, Brazil).

Low-density lipoprotein (LDL) concentration was calculated using the Friedewald formula: [LDL] = [total cholesterol] − [HDL] − [triglycerides ÷ 5] [19]. Glycated hemoglobin (HbA1c), C-reactive protein (CRP) levels, were determined by an immunoturbidimetry technique (Lab-Max 240, Labtest, Lagoa Santa, MG, Brazil) using commercial kits (Labtest, Lagoa Santa, MG, Brazil).

For the classification of dyslipidemia, the criteria were established according to the Guidelines of the Brazilian Diabetes Society (2019): Total Cholesterol (<190mg / dL), LDL-C (<130mg / dL), HDL-C (≤40mg / dL) and Triglycerides (≥150mg / dL) [20]. For the values of Hba1c (≤6,5%), total cholesterol (≤190mg / dL), HDL (≥40mg / dL) and LDL (<130mg / dL), the cutoff points adopted followed the recommendations of the Guidelines of the Brazilian Diabetes Society (2019) [20]. For Triglycerides (<150mg / dL), based on the Brazilian Archives of Cardiology (2017) [21]. Serum creatinine according to sex, (women ≤0.995mg / dL; men ≤1.20mg / dL), according to the Brazilian Society of Nephrology (2011) [22]. C Reactive Protein (≤3mg / dL) according to the Brazilian Guidelines on Dyslipidemias and Atherosclerosis Prevention [23], and for Malondialdehyde, the reference value adopted was (≤3.31μM) [24].

The anthropometric variables were body weight (kg), which was measured with the use of a scale, and height (cm). The body mass index (BMI) was calculated by dividing body weight by height squared (in meters) and the individuals were classified according to the presence of overweight and obesity, according to Body Mass Index (BMI) in kg / m$^2$. The values for the adult age were: overweight when BMI = 25–29.9 kg/m2 and obesity when BMI ≥ 30kg /m2. For the elderly patients, overweight was defined as BMI = ≥27 kg / m$^2$ [25]. Clinical and anthropometric information such as gender, age, T2DM time, BMI and blood pressure were obtained in the clinical evaluation by a nutritionist from the research team and by the team in the endocrinology service.

## 2.4. Isolation of leukocyte DNA

To obtain leukocyte DNA, appropriate protocols were used. The samples were diluted in an initial lysis solution containing 10mM Tris-HCl pH, 8.5mM EDTA, 0.3M sucrose and 1% Triton-X-100. Centrifugation was performed at 3.200 rpm, and the supernatant was discarded. This process was repeated 3 times to obtain a leukocyte precipitate free from hemoglobin remnants. The precipitate was resuspended in a lysis solution containing 10mM Tris-HCl pH8.0, 0.5% sodium dodecyl sulfate (SDS), 5mM EDTA and 0.2μg proteinase K (Invitrogen, Carlsbad, CA, USA) and incubated at 55˚C in a water bath for 7h. Then, 500μl of an aqueous solution of 1mM EDTA and 7.5M ammonium acetate was added and mixed for 30 seconds. The mixture was centrifuged for 10min at 14.000g at 4˚C, and 700μl of the supernatant was transferred to a new tube where DNA precipitation was performed with 540μl of iced isopropanol. The DNA precipitate was washed with 70% ethanol, centrifuged (12.000g for 5min), dried and resuspended in Tris-EDTA buffer pH 8.0 [26]. The samples were kept at -20˚ C until genetic analysis.

## 2.5. rs1544410 genotyping

Genotypes were determined by PCR-RFLP. Appropriate primers were used to amplify the region of the gene containing the polymorphism [27], and amplification occurred under the following conditions: denaturation 94˚ C for 5 minutes, 30 cycles of denaturation (1 minute at 94˚ C), annealing (1 minute at 58˚ C) and extension (3 minutes at 72˚ C) with an extra 10 minute extension step minutes. The 825 bp product was digested with BsmI which recognizes and cleaves the polymorphic allele (b) generating two fragments (650 bp and another 175 bp) while the wild allele (B) remains at 825 bp. The genotypes were analyzed by 15% polyacrylamide gel electrophoresis and 0.5% silver nitrate staining.

## 2.6. Statistical analysis

SPSS 26.0 software (SPSS Inc., Chicago, IL, USA) was used for statistical analysis. Normality in continuous variables was assessed using the Kolmogorov-Smirnov test, and distributions with $p > 0.05$ were accepted as variables with normal distribution and expressed as mean ± standard deviation values and evaluated by the t test of independent samples. The 'non normally distributed' were expressed as median values and 95% confidence intervals and compared using the Mann-Whitney U test between groups. Nominal categorical variables were expressed as total number and percentage, analyzed by chi-square. Hardy-Weinberg equilibria were calculated to assess expected and observed genotypic and allelic frequencies were tested by chi-square. Data from the RD group were analyzed according to rs1544410 genotypes by independent T test and its non-parametric counterpart, and then a logistic regression model was used to establish which variables (fasting glucose, insulin therapy, sex, age and time of DM2) could be influenced by rs1544410 genotypes. A p value $< 0.05$ was considered significant for all analyses.

## 3. Results

The control (DM) and RD groups were similar regarding the frequency of hypertension, dyslipidemia, family history of DM, smoking, sedentary lifestyle, age, HbA1c, fasting glucose, LDL-c, triglycerides and BMI, however, in the group with DR, male gender (p = 0.001), insulin therapy (p = 0.000), and serum creatinine (p = 0.000) were higher in addition to a longer duration of DM2 (p = 0.000). In the DM group, patients had higher values of total cholesterol (p = 0.017) and HDL-c fraction (p = 0.008) (Table 1).

**Table 1. Clinical, biochemical and metabolic parameters of the studied sample.**

| | DM (128) | RD (157) | p-value OR (IC 95%) |
|---|---|---|---|
| **Sex (M%)** | 31 (24.2%) | 68 (43.6%) | **0.001*** 0.414 (0.247–0.691) |
| **Hypertension (%)** | 82 (65.1%) | 116 (74.4%) | 0.090 0.643 (0.385–1.074) |
| **Dyslipidemia (%)** | 102 (79.7%) | 126 (80.3%) | 0.905 0.965 (0.539–1.729) |
| **Family history of DM (%)** | 87(71.3%) | 110(72.8%) | 0.778 0.926 (0.545–1.576) |
| **Insulin treatment (%)** | 37 (29.4%) | 112 (71.8%) | **0.000*** 0.163 (0.097–0.274) |
| **Tabagism (%)** | 07 (5.5%) | 08 (5.1%) | 0.886 1.079 (0.380–3.061) |
| **Sedentary lifestyle (%)** | 68(53.5%) | 81 (51.9%) | 0.786 1.067 (0.667–1.706) |
| **Age (years)** | 59.2 ± 10.1 | 60.8 ±8.3 | 0.159 |
| **DM2 duration (years)** | 7 (7.6–9.1) | 17 (15–17.5) | **0.000*** |
| **HbA1c (%)** | 7.9±1.6 | 7.8±1.4 | 0.305 |
| **Glucose (mg/dL)** | 152.4±50.7 | 141.1±52 | 0.066 |
| **Total cholesterol (mg/dL)** | 183.4 ±41.1 | 170.9 ±45.6 | **0.017*** |
| **HDL (mg/dL)** | 46.4 ±12.4 | 42.7±11.1 | **0.008*** |
| **LDL (mg/dL)** | 104.2±37.5 | 98.1±39 | 0.185 |
| **Triglycerides (mg/dL)** | 163.5 (154.38–188.6) | 115 (125.11–157) | 0.094 |
| **Serum creatinine (mg/dL)** | 0.71 (0.71–0.80) | 0.84 (0.85–0.95) | **0.000*** |
| **BMI (kg/m$^2$)** | 29.7 ±5.7 | 29 ±5.1 | 0.282 |

*p <0.05 = statistically significant difference.

Categorical variables analyzed by chi-square test, values expressed as total value (and percentage) and confidence interval. Quantitative variables analyzed by independent t test (when in normal distribution and/or homogeneity) and presented as mean and ±SD; Variables that did not show normal distribution and/or homogeneity were analyzed using the Mann-Whitney U test and represented by median and (95%CI).

Table 2 shows the distributions of genotypes (BB + Bb / bb; Bb + bb / BB; BB + bb / Bb) and alleles (B / b) as a function of the DM control group compared to different DR categorizations; DR with ≤10 years of DM2 diagnosis; DR with >10 years of DM2 diagnosis; groups with RDNP and RDP, respectively. In all groups there was no difference in the genotypic or allelic distribution of the groups. The minor allele (B) frequency (MAF) was 0.46 in control DM group and 0.43 in the DM RD group.

Clinical parameters of the RD group (n = 157) were analyzed according to BsmI genotypes (BB x Bb+ bb). In the group with the Bb+bb genotype, insulin therapy was significantly more frequent (p = 0.014) and fasting glucose had a higher mean value in the same group (p = 0.047) (Table 3). A logistic regression model, adopting the BsmI genotypes as the dependent variable and with correction/adjustment variables, revealed that fasting glucose (p = 0.018) and insulin therapy (p = 0.036) were statistically significant predictors of increased odds of risk in the group of patients with the Bb+bb genotype (Table 4). The glycemia difference between the genotypic groups presents a Cohen's effect size d = 0.5, both comparing only the BB x bb group and Bb+bb x BB. According to the literature, this value means an effect of medium magnitude [28].

## 4. Discussion

Vitamin D in its active form acts through its specific receptor (*VDR*), expressed in human tissues including the retina [29]. In addition to the association of lower levels of Vitamin D and increased risk of DM2 and RD [8, 13], several studies have also related *VDR* gene polymorphisms as risk factors for DM2 [30], pathogenesis [14, 15, 31], and DR progression [32].

**Table 2. Genotypic and allelic distribution of the polymorphism rs1544410 among type 2 diabetic patients without DR and RD, <10 years DM diagnostic, ≥10 years DM diagnostic, RDNP and RDP.**

| rs1544410 (BsmI) | | | |
|---|---|---|---|
| Genotypes/Alleles | DM (128) | DR (157) | p-value OR (IC 95%) |
| BB+Bb / bb | 65.63% (84) / 34.37% (44) | 61.78% (97) / 38.22% (60) | 0.585 1.181 (0.7259–1.921) |
| Bb+bb / BB | 72.66% (93) / 27.34% (35) | 75.16% (118) / 24.84% (39) | 0.731 0.8782 (0.5163–1.494) |
| BB+bb / Bb | 61.72% (79) / 38.28% (49) | 63.06% (99) / 36.94% (58) | 0.913 0.9445 (0.5834–1.529) |
| B /b | 46.48% (119) / 53.52% (137) | 43.31% (136) / 56.69% (178) | 0.501 1.137 (0.8157–1.584) |
| Genotypes/Alleles | DM (128) | DR (49) (≤10 years DM diagnosis) | p OR (IC 95%) |
| BB+Bb / bb | 65.63% (84) / 34.37% (44) | 63.27% (31) / 36.73% (18) | 0.906 1.109 (0.5582–2.201) |
| Bb+bb / BB | 72.66% (93) / 27.34% (35) | 73.47% (36) / 26.53% (13) | 0.913 0.959 (0.4560–2.019) |
| BB+bb / Bb | 61.72% (79) / 38.28% (49) | 63.27% (31) / 36.73% (18) | 0.987 0.936 (0.4735–1.851) |
| B /b | 46.48% (119) / 53.52% (137) | 44.90% (44) / 55.10% (54) | 0.8817 1.066 (0.6677–1.702) |
| Genotypes/ Alleles | DM (128) | DR (108) (>10 years DM diagnosis) | p-value OR (IC 95%) |
| BB+Bb / bb | 65.63% (84) / 34.37% (44) | 61.11% (66) / 38.89% (42) | 0.561 1.215 (0.7138–2.068) |
| Bb+bb / BB | 72.66% (93) / 27.34% (35) | 75.93% (82) / 24.07% (26) | 0.673 0.8425 (0.4679–1.517) |
| BB+bb / Bb | 61.72% (79) / 38.28% (49) | 62.96% (68) / 37.04% (40) | 0.951 0.9484 (0.5589–1.609) |
| B / b | 46.48% (119) / 53.52% (137) | 42.59% (92) / 57.41% (124) | 0.451 1.171 (0.8128–1.686) |
| Genotypes/ Alleles | DM (128) | DRNP (103) | p-value OR (IC 95%) |
| BB+Bb / bb | 65.63% (84) / 34.37% (44) | 61.17% (63) / 38.83% (40) | 0.574 1.212 (0.7073–2.077) |
| Bb+bb / BB | 72.66% (93) / 27.34% (35) | 76.70% (79) / 23.30% (24) | 0.583 0.8072 (0.4430–1.471) |
| BB+bb / Bb | 61.72% (79) / 38.28% (49) | 62.14% (64) / 37.86% (39) | 0.943 0.9825 (0.5757–1.677) |
| B / b | 46.48% (119) / 53.52% (137) | 42.23% (87) / 57.77% (119) | 0.412 1.188 (0.8207–1.720) |
| Genotypes/Alleles | DM (128) | DRP (54) | p-value OR (IC 95%) |
| BB+Bb / bb | 65.63% (84) / 34.37% (44) | 62.96% (34) / 37.04% (20) | 0.862 1.123 (0.5792–2.177) |
| Bb+bb / BB | 72.66% (93) / 27.34% (35) | 72.22% (39) / 27.78% (15) | 0.9152 1.022 (0.5818–2.081) |
| BB+bb / Bb | 61.72% (79) / 38.28% (49) | 64.81% (35) / 35.19% (19) | 0.821 0.8752 (0.4511–1.698) |
| B / b | 46.48% (119) / 53.52% (137) | 45.37% (49) / 54.63% (59) | 0.937 1.046 (0.6658–1.643) |

Statistical analysis performed with the Chi-square test. *VDR*, vitamin D receptor.

The relationship of polymorphisms with the occurrence of chronic complications in diabetics has also been explored [33–35] and has a pronounced importance in the early identification of individuals at higher risk of developing specific complications. The SNP polymorphism rs1544410 is located in intron 8 in the 3' regulatory region, which is involved in gene expression, especially through the regulation of messenger RNA stability [36, 37], which can result in reduced expression of the *VDR* gene [38]. Another possible mechanism is that the alternation of BsmI in the intronic sequence may influence protein expression [39].

**Table 3. Clinical, biochemical and metabolic parameters of the RD group according to rs1544410 genotypes.**

| | BB (n = 39) | Bb+bb (n = 118) | p-value OR (IC 95%) |
|---|---|---|---|
| **Sex (M%)** | 20 (51.3%) | 48 (41%) | 0.263 |
| | | | 1.513 (0.731–3.134) |
| **Hypertension (%)** | 26 (66.7%) | 90 (76.9) | 0.204 |
| | | | 0.600 (0.272–1.325) |
| **Dyslipidemia (%)** | 08 (29.6%) | 25 (33.3%) | 0.724 |
| | | | 0.842 (0.324–2.189) |
| **Family history of DM (%)** | 24 (63.2%) | 86 (76.1%) | 0.121 |
| | | | 0.538 (0.245–1.184) |
| **Insulin treatment (%)** | 22 (52.4%) | 90 (76.9%) | **0.014**[*] |
| | | | 0.388 (0.181–0.834) |
| **Tabagism (%)** | 02 (5.1%) | 06 (5.1%) | 1.000 |
| | | | 1.000 (0.193–5.171) |
| **Sedentary lifestyle (%)** | 20 (51.3%) | 61 (52.1%) | 0.926 |
| | | | 0.966 (0.468–1.996) |
| **Age (years)** | 60.6±8.9 | 60.8±8.1 | 0.926 |
| | | | 1.533 (-3.171–2.886) |
| **DM2 duration (Years)** | 15 (13.2–18.6) | 18 (15–17.7) | 0.640 |
| **HbA1c (%)** | 7.6±1.3 | 7.7±1.5 | 0.593 |
| **Glucose (mg/dL)** | 126.3±44 | 145.8±54.7 | **0.047**[*] |
| **Total cholesterol(mg/dL)** | 176.1±59 | 171.4±46.8 | 0.656 |
| **HDL** | 41 (38.6–46.9) | 42 (40.6–44.4) | 0.855 |
| **LDL** | 87.8 (90.2–104.6) | 100 (89–124.5) | 0.331 |
| **Triglycerides (mg/dL)** | 115 (112.4–158.2) | 124 (136.4–169.5) | 0.257 |
| **Creatinine (mg/dL)** | 0.83 (0.81–1.0) | 0.84 (0.84–0.95) | 0.677 |
| **BMI (KG/M$^2$)** | 27.5±4.6 | 28.7±4.6 | 0.163 |

[*] p <0.05 = statistically significant difference.

Categorical variables analyzed by chi-square test, values expressed as total value (and percentage) and confidence interval. Quantitative variables analyzed by independent t test (when in normal distribution and/or homogeneity) and presented as mean and ±SD; Variables that did not show normal distribution and/or homogeneity were analyzed using the Mann-Whitney U test and represented by median e (95%CI).

The MAF found in this study was 0.47 in the diabetic group and 0.43 in DR patients while in another Brazilian sample from Minas Gerais State (Southeast region) of the country, the MAF found was 0.40 in the group of type 2 diabetics. In the samples of the 1000 genome project, obtained from healthy population, the MAF (A) for the SNP is 0.29. According to the SNP

**Table 4. Logistic regression model analyzing the influence of rs1544410 genotypes and metabolic variables in patients with DR.**

| | B | p-value OR (IC 95%) |
|---|---|---|
| **Glucose** | 0,010±0,004 | **0.018**[*] 1,010 (1,002–1.018) |
| **Insulin treatment** | 0.982±0.470 | **0.036**[*] 2.671 (1.064–6.705) |
| **Sex** | 0.388±0.409 | 0.342 1.475 (0.662–3.286) |
| **Age (years)** | 0.017±0.025 | 0.498 1.017 (0.968–1.069) |
| **DM2 duration (Years)** | -0,010±0.028 | 0.731 0.990 (0.937–1.047) |

Source: research data. Binary Logistic Regression. Dependent variable: BB x Bb+bb (Bb+bb reference);

[*]p>0.05;

B = beta regression coefficient ± standard error; OR = upper and lower odds limits; Insulin treatment (use of insulin as a reference); Sex (male as reference).

database (ncbi.nlm.gov) in two different Latin American populations the MAF found was 0.36 (n = 798) and 0.23 (n = 3896), in Europeans it was 0.39 and in African Americans 0.26. Interestingly, in Asian populations the MAF is significantly lower, showing that it is a SNP that is highly influenced by ethnic aspects of the population under analysis. These informations evidenciate similarity between the MAF found in this study and another sample of Brazilian diabetics and a Latin American sample.

The genotypic distribution found deviated from the Hardy Weinberg Equilibrium in both groups, DM2 and DM2 with DR. This finding has not been uncommon in studies involving the polymorphism in question and diabetes. The HWE imbalance was present in a study conducted in the USA [40] and another in Chile [41]. In a recent meta-analysis that evaluated the effect of rs1544410 on the risk of DM2, of the 37 studies analyzed, 10 showed a deviation from the HWE in samples from different locations around the world [42].

The "b" allele was associated with lower HDL levels and increased risk for T2DM and obesity in Arabs [43], and the occurrence of DR in Croatian population with DM1 [32] the bb genotype conferred greater insulin resistance in Caucasians [40].

Despite the evidence, there are few studies exploring the association of the SNP rs1544410 and RD; one of them in DM1 [32] and four in DM2 [14, 15, 44, 45]. In these studies, one found an association of the B allele with lower risk of DR in DM2 [14] and two studies found association between RD and the bb genotype in DM1 [32] and DM2 [15], whereas in other studies no genotypic or allelic association was found [44, 45]. The geographic regions involved in these results cover Europe [32, 44] and Asia [14, 15, 45], however there are no studies analyzing the relationship in the Latin American population, which characterizes this study as pioneer. The results of the present study do not show an association of the rs1544410 polymorphism with DR in any of the studied scenarios, either by classifying patients by duration of diabetes, by DR stage or by simple diagnosis of the condition.

The time of diabetes diagnosis was also evaluated by Búcan (2009), Hong (2015), Zhong (2015) and Cyganek (2006), and the results found reinforce the data obtained in the study presented here. A single study [14] used the RD stage classification variable controlling the time since DM2 diagnosis (mean of 13 years) between the groups. In this study, the BB, Bb and the B allele genotypes were related to a lower risk of DR.

Meta-analyses [17, 29] point to the absence of association between rs1544410 and DR in both DM1 and DM2. However, the meta-analysis conducted by Song (2019) indicates an association restricted to the Indian population, but that is the result of a single study [15]. Therefore, we consider it important to explore this possibility in the Brazilian population. As the association did not result in positive data, we decided to investigate the effect of different rs1544410 genotypes in patients with DR on clinical and metabolic parameters.

Among the existing studies with DM2 and DR, only one controlled the duration of DM2 between the groups, with no statistically significant differences [14], while Zhong (2015), Cyganek (2006) and Ezhilarasi (2018) present difference or do not report the comparison. Aware of the great influence of this variable on the studied aspect, in this work, we performed different comparison scenarios, including separating patients with DR up to 10 years of diagnosis and more than 10 years of DM2 diagnosis, comparing genotypic and allelic distributions with the control group. In none of these scenarios were the genotypes or alleles statistically different from the control.

As the association analysis found conflitant results in comparison to some previous studies [14, 15], we decided to investigate the effect of different rs1544410 genotypes on metabolic parameters, in order to assess whether despite not impacting the risk of DR, having knowledge of the genotype would have an impact on the knowledge of the individual's metabolic profile. We compared the genotypic groups within the profile of DR carriers and identified that

carriers of the b allele (Bb and bb) had higher blood glucose levels and a greater chance of insulin therapy than carriers of the BB genotype.

Persistent hyperglycemia triggers most of the processes that promote the development of DR and high blood glucose concentrations and high levels of HbA1C characterize poor glycemic control and associate them with an independent risk factor for DR [5, 6, 46–49].

The association between *VDR* variants (BsmI) and insulin resistance-related diseases, demonstrate controversial conclusions. While the BB+Bb genotypes and the B allele were related to insulin resistance-related diseases in darkly pigmented Caucasians [50], in a prospective cohort, the B allele was associated with increased insulin secretion in women with a history of gestational DM [51]. The BsmI polymorphism of the *VDR* gene was also associated with glycemic outcomes in other populations. BsmI was tested in a homogeneous population of healthy young men, at an age before the typical onset of DM2, in which physical activity was also measured. Individuals with the BB genotype had significantly higher fasting glucose levels than individuals with the Bb or bb genotype. This finding was associated with low physical activity (≤3h per week), but the effect was absent in men with a high degree of physical activity [52].

In addition to associations related to the vitamin D receptor gene and BsmI genotypes with glycemic markers, low serum concentrations of 25OHD3 have been associated with reduced insulin sensitivity, impaired glucose metabolism, and metabolic syndrome [53, 54]. The mechanism by which vitamin D affects insulin sensitivity is still unknown. Vitamin D can stimulate the expression of insulin receptors in peripheral tissues and, thus, increase glucose transport [55]. Insulin-mediated glucose uptake is calcium dependent and therefore vitamin D status may indirectly influence glucose uptake [56].

The b allele was considered a risk factor for low serum concentrations of 25-hydroxyvitamin D, in addition the same study reports that BsmI influenced glucose and HOMA-IR concentrations even after adjustments in Brazilian (Amazon State) children [57].

These evidences demonstrate the importance of going beyond the simple genotype-clinical condition relationship. It is possible that the lack of association found in some populations, including this study, is a reflection of the applied experimental design, and the influencing effect of other aspects not measured or analyzed, including other SNPs. The association between higher glucose levels and the b allele is relevant in the context studied, as this same genetic characteristic seems to influence vitamin D levels. The results found here encourage further analysis, taking into account clinical, metabolic and vitamin D levels of patients when evaluating the association of VDR genotypes and DR.

This is the first study to analyze a Latin American population for the influence of rs1544410 in the context of DR. The study has limitations related to its cross-sectional nature. The results found here encourage the performance of studies with other populations and with different experimental models to unravel the effects of other important components for DR.

## 5. Conclusion

We did not identify the BsmI SNP (rs1544410) of the *VDR* gene as a risk factor for DR, but the b allele may be indicative of increased glycemic levels and insulinoterapy in the population with DR. The relationship between *VDR* BsmI and susceptibility to DR in a Latin American population should be further studied.

## Supporting information

**S1 Data.**
(XLSX)

## Acknowledgments

We thank at the Lauro Wanderley University Hospital, Federal University of Paraíba and all participants in the study for participation and the financial support institutions, National Council for Scientific and Technological Development (CNPq, Brasilia, Brazil), the Coordination for the Improvement of Higher Education Personnel (CAPES, Brasília, Brazil) and the State of Paraíba Research Support Foundation (FAPESQ, Paraíba, Brazil). The authors declare that they have no competing interests.

## Author Contributions

**Conceptualization:** Darlene Camati Persuhn.

**Formal analysis:** Caroline Severo de Assis, Alexandre Sérgio Silva, Rafaela Lira Formiga Cavalcanti de Lima.

**Funding acquisition:** Darlene Camati Persuhn.

**Investigation:** Caroline Severo de Assis, Tainá Gomes Diniz, João Otávio Scarano Alcântara, Vanessa Polyana Alves de Sousa Brito, Rayner Anderson Ferreira do Nascimento, Mayara Karla dos Santos Nunes, Isabella Wanderley de Queiroga Evangelista, Marina Gonçalves Monteiro Viturino.

**Methodology:** Darlene Camati Persuhn.

**Project administration:** Darlene Camati Persuhn.

**Supervision:** Darlene Camati Persuhn.

**Writing – original draft:** Caroline Severo de Assis.

**Writing – review & editing:** Darlene Camati Persuhn.

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
