## [Decision Letter · Decision Letter 0]

25 Oct 2021

PONE-D-21-31455Metabolic impact of the VDR rs1544410 in diabetic retinopathyPLOS ONE

Dear Dr. ASSIS,

Thank you for submitting your manuscript to PLOS ONE. After careful consideration, we feel that it has merit but does not fully meet PLOS ONE’s publication criteria as it currently stands. Therefore, we invite you to submit a revised version of the manuscript that addresses the points raised during the review process.

We look forward to receiving your revised manuscript.

Kind regards,

Kanhaiya Singh, Ph.D

Academic Editor

PLOS ONE

Journal Requirements:

Additional Editor Comments:

Please address the reviewers concern about the sample size and power of the study.

Reviewers' comments:

Reviewer's Responses to Questions

**Comments to the Author**

1. Is the manuscript technically sound, and do the data support the conclusions?

Reviewer #1: No

Reviewer #2: Partly

2. Has the statistical analysis been performed appropriately and rigorously? 

Reviewer #1: I Don't Know

Reviewer #2: Yes

3. Have the authors made all data underlying the findings in their manuscript fully available?

Reviewer #1: Yes

Reviewer #2: No

4. Is the manuscript presented in an intelligible fashion and written in standard English?

Reviewer #1: Yes

Reviewer #2: Yes

5. Review Comments to the Author

Reviewer #1: The present manuscript by de Assis et al. describes case control association study to investigate association of BsmI and clinical parameters in diabetic retinopathy in a DM2 population in Northeastern Brazil. The authors have conducted clinical, biochemical, and anthropometric analysis, and found no association between rs1544410 and the presence of DR; however, they should provide justifications for following points:

1. The study is based on a small sample size of only 176 subjects (100 patients and 76 controls). Have authors considered power calculation and estimated required sample size?

2. Why have authors not considered healthy individuals as controls? Clinical characterization section mentions that DM2 patients without DR have been taken as control. DM2 patients currently not diagnosed with DR may develop it in future because risk of diabetic retinopathy is always associated with uncontrolled diabetes in long-term.

3. Why have authors included familial cases of DM (as indicated in Table 1) in case control studies? Should it not be based on sporadic cases only? Further, cases of other systemic problems such as hypertension have also been included. Such selection criteria of subjects might not represent true analysis of a complex disease such as diabetic retinopathy.

The authors should take necessary steps to address the above-mentioned points before reaching a conclusion.

Reviewer #2: The study by Severo de Assis et al. has investigated the association between BsmI

polymorphism (rs1544410) of the VDR gene and clinical parameters in diabetic retinopathy in a North-eastern Brazilian cohort consisting of Diabetes mellitus type-2 (DM2) patients. Overall, this is a clear, concise, and well-written manuscript. The introduction is relevant and the discussion section is enriched with sufficient information about the previous study findings for readers to follow the present study rationale and procedures. The methods are appropriate, results are clear. The study has not detected any significant association between the rs1544410 and causation of diabetic retinopathy (DR). However following points should be taken into consideration in the present form of this manuscript.

Major comments

1. In section ‘rs1544410 genotyping’ the RFLP method used for genotyping looks a bit archaic. Incomplete digestion can lead to erroneous genotypes. The authors should confirm genotypes with another method (i.e TaqMan or Sanger sequencing) to demonstrate that the bands they observed in gel represented actual genotypes and there were no errors due to incomplete digestion (i.e. failure to cut completely or failure to cut at all due to technical problems being interpreted erroneously as an absence of cut sites).

2. The authors should investigate rs1544410 based haplotype frequencies are associated with the presence of diabetic retinopathy in the conditions studied. Authors can use freely available computational tools such as SHEsis Online haplotype analysis software for this purpose.

3. The size of cohort is small. The authors should include Genetic power calculation for estimating the sample size in this study.

4. The authors should elaborate the exclusion criteria which were followed during recruitment of patients in the cohort.

5. The authors should include the allelic frequencies of SNP rs1544410 by making a query in the 1000 genomes database (https://www.internationalgenome.org), Single Nucleotide Polymorphism database (dbSNP, http://www.ncbi.nlm.nih.gov/SNP) and ClinVar (http://www.ncbi.nlm.nih.gov/clinvar/) database.

Minor comments

Please recheck the figure legends and statistical tables thoroughly for language errors.

Overall, results of the present study do not show an association of the rs1544410 polymorphism with the risk of DR causation. However, the authors have found statistically significant association with a biochemical parameter; fasting blood glucose of the DR group. In my opinion, the authors should significantly enrich the results of this manuscript by increasing the sample size and include Cox proportional hazards regression model or logistic regression models to analyse the effect of different risk factors on the causation of DR. In current study, the authors can also include a systematic and detailed meta-analysis to further enrich this manuscript.

6. PLOS authors have the option to publish the peer review history of their article (what does this mean?). If published, this will include your full peer review and any attached files.

Reviewer #1: No

Reviewer #2: No

---

## [Author Response · Author response to Decision Letter 0]

4 Dec 2021

1. Is the manuscript technically sound, and do the data support the conclusions?

Reviewer #1: No

Reviewer #2: Partly

R: We added samples to the sample in both experimental groups and used other statistical resources (logistic regression) to demonstrate the consistency of the relationship found between fasting glucose and insulin use with the b allele of the studied polymorphism.

2. Has the statistical analysis been performed appropriately and rigorously?

Reviewer #1: I Don't Know

Reviewer #2: Yes

R: We hope that we have presented well-founded statistical analyzes in the new version of the article.

3. Have the authors made all data underlying the findings in their manuscript fully available?

Reviewer #1: Yes

Reviewer #2: No

R: We are providing the complete database.

4. Is the manuscript presented in an intelligible fashion and written in standard English?

Reviewer #1: Yes

Reviewer #2: Yes

5. Review Comments to the Author

Reviewer #1: The present manuscript by de Assis et al. describes case control association study to investigate association of BsmI and clinical parameters in diabetic retinopathy in a DM2 population in Northeastern Brazil. The authors have conducted clinical, biochemical, and anthropometric analysis, and found no association between rs1544410 and the presence of DR; however, they should provide justifications for following points:

1. The study is based on a small sample size of only 176 subjects (100 patients and 76 controls). Have authors considered power calculation and estimated required sample size?

R: We agree with the reviewer when he indicated that a case-control study consisting of 176 individuals is small. To improve this aspect, we increased the number of groups to 128 in the control group and 157 in the RD group (total = 285).

Taking into account the genotypic distribution found in this study, OR BB x Bb+bb= 1.13, disease frequency= 0.2; MAF = 0.44; significance level 0.05, it would take ~2000 patients divided between cases and controls to achieve an expected power of ~0.8. This would be the number necessary to discriminate a possible difference between cases and controls due to the high allelic frequency and the OR of the genotypic distribution being very close to 1. This number of patients is unfeasible for our recruitment pace. However, in the review published by Song et al (2019), 7 studies compared rs1544410 genotypes in groups of DM x DM patients with DR. In only one of these studies the number of patients with DR was greater than what is presented in this article. Therefore, we consider that the study aggregates by bringing information from a population not yet studied in the literature.

It is also necessary to consider that the main finding of this study is not the lack of association of polymorphism with DR, but the differences in blood glucose between the genotypes.

Song, N., Yang, S., Wang, Y., Tang, S., Zhu, Y., Dai, Q., & Zhang, H. (2019). The Impact of Vitamin D Receptor Gene Polymorphisms on the Susceptibility of Diabetic Vascular Complications: A Meta-Analysis. Genetic Testing and Molecular Biomarkers, 23(8), 533–556. doi:10.1089/gtmb.2019.0037 

2. Why have authors not considered healthy individuals as controls? Clinical characterization section mentions that DM2 patients without DR have been taken as control. DM2 patients currently not diagnosed with DR may develop it in future because risk of diabetic retinopathy is always associated with uncontrolled diabetes in long-term.

R: It is true that patients in the control group can develop the complication over time, so we stratified the patient groups in genotypic comparisons not only by the degree of complication (RDNP and RDP) but also by the time of diabetes diagnosis (greater and less than 10 years). In all cases, the distribution of genotypes and alleles was similar, without any trend towards differences between groups. This reinforces the hypothesis that the polymorphism does not influence the occurrence of the complication in the studied group.

The inclusion of a control group without DM is performed in some studies. However, the direct comparison between a group without DM with DM with DR is not as informative as identifying differences when the comparison group consists only of diabetics without the complication. This is because the main interest is to identify risk markers for the complication in diabetes carriers. The advantage of including samples without DM in this study would be to bring a representative group of genotypic distribution in the absence of the pathology. It was not possible to fulfill this reviewer's request because this was not foreseen in the objectives of the original project.

3. Why have authors included familial cases of DM (as indicated in Table 1) in case control studies? Should it not be based on sporadic cases only? Further, cases of other systemic problems such as hypertension have also been included. Such selection criteria of subjects might not represent true analysis of a complex disease such as diabetic retinopathy.

The authors should take necessary steps to address the above-mentioned points before reaching a conclusion.

R: Patients included in the study were not related. The term “Family DM” was mistakenly used and replaced by the “family history of DM”. In this item, we investigated patients who claimed to have at least one relative up to the second degree with DM in their family.

RD is not an isolated clinical condition and affects individuals who usually have more than 5 years of diabetes. One of its main risk factors is high blood pressure, control of glycemic levels and dyslipidemia. More than 70% of patients with DR have SAH, which would make it very difficult to exclude patients with this and other chronic conditions from the sample. These data are in the descriptive presentation of all articles comparing diabetic patients with DM with DR in relation to specific genotypes.

As can be seen in table 3, only blood glucose and insulinoterapy were different when comparing the genotypic groups, although when comparing the clinical groups (Table 1) it is possible to identify other variables that distinguish the patients (duration of diabetes, creatinine, lipid parameters and %male sex). Such variables were analyzed and proved not to be influenced by genotype variation in the study population.

Reviewer #2: The study by Severo de Assis et al. has investigated the association between BsmI

polymorphism (rs1544410) of the VDR gene and clinical parameters in diabetic retinopathy in a North-eastern Brazilian cohort consisting of Diabetes mellitus type-2 (DM2) patients. Overall, this is a clear, concise, and well-written manuscript. The introduction is relevant and the discussion section is enriched with sufficient information about the previous study findings for readers to follow the present study rationale and procedures. The methods are appropriate, results are clear. The study has not detected any significant association between the rs1544410 and causation of diabetic retinopathy (DR). However following points should be taken into consideration in the present form of this manuscript.

Major comments

1. In section ‘rs1544410 genotyping’ the RFLP method used for genotyping looks a bit archaic. Incomplete digestion can lead to erroneous genotypes. The authors should confirm genotypes with another method (i.e TaqMan or Sanger sequencing) to demonstrate that the bands they observed in gel represented actual genotypes and there were no errors due to incomplete digestion (i.e. failure to cut completely or failure to cut at all due to technical problems being interpreted erroneously as an absence of cut sites).

R: We appreciate the reviewer's analysis and agree that methods involving sequencing or probes are more recent. But it is possible to obtain genotypic quality data working manually using the following features: positive, negative and blank controls in all runs, guarantee of complete digestion by always repeating heterozygotes and doing random and blind repetition of at least one tenth of the samples analyzed in the study. In our laboratory, we do not have resources for acquiring probes, the expiration time also makes their use in our projects unfeasible. Therefore, our available method is RFLP. In order to demonstrate the data quality with which we work, I share below an image of one of our recent electrophoresis from the work under evaluation:

 The 875 and 650pb fragments are distant and allow a clear differentiation between the genotypes. As a guarantee feature of the BB genotype, we have the absence of the 175bp fragment. The spot, which appears close to 400bp, corresponds to the digestion reaction mixture, probably the restriction endonuclease. As it runs well away from the genotypic decision bands, it also does not cause any interference in the interpretation of the result.

2. The authors should investigate rs1544410 based haplotype frequencies are associated with the presence of diabetic retinopathy in the conditions studied. Authors can use freely available computational tools such as SHEsis Online haplotype analysis software for this purpose.

R: We found difficulties in accessing the program in question, although it remains used in articles published in 2021. However, we found that its main use is linked to haplotype analyses. The VDR gene has many polymorphisms, but the main ones are FokI, TaqI, ApaI and BsmI (studied in this article). As we did not study the other polymorphisms that could compose the haplotypes, it would not be possible to use this tool in our data, we supose. There are also few studies that assess the haplotypes in relation to the risk of DR, one of them being in Han Chinese type 2 diabetes patients [47] which found an association between the FokI and DR polymorphism and also identified a frequent haplotype in the DR group, which includes the b allele of the BsmI – which reinforces data from other studies that this would be the risk allele for the condition under study. As the study approach is specific to the BsmI polymorphism, we decided not to incorporate such information into the text of the article.

Zhong X, Du Y, Lei Y, Liu N, Guo Y, Pan T. Effects of vitamin D receptor gene polymorphism and clinical characteristics on risk of diabetic retinopathy in Han Chinese type 2 diabetes patients. Gene. 2015 Jul 25;566(2):212-6. doi: 10.1016/j.gene.2015.04.045. 

3. The size of cohort is small. The authors should include Genetic power calculation for estimating the sample size in this study.

R: We agree with the reviewer when he indicated that a case-control study consisting of 176 individuals is small. Thinking about improving this aspect of the work, we increased the number of groups to 128 in the control group and 157 in the RD group (total = 285).

Taking into account the genotypic distribution found in this study, OR BB x Bb+bb= 1.13, disease frequency= 0.2; MAF = 0.44; significance level 0.05, it would take ~2000 patients divided between cases and controls to achieve an expected power of ~0.8. This would be the number necessary to discriminate a possible difference between cases and controls due to the high allelic frequency and the OR of the genotypic distribution being very close to 1. This number of patients is unfeasible for our recruitment pace and also for our structure. However, in the review published by Song et al (2019), 7 studies compared rs1544410 genotypes in groups of DM x DM patients with DR. In only one of these studies the number of patients with DR is greater than what is presented in this article. Therefore, we consider that the study aggregates by bringing information from a population not yet studied in the literature.

Regarding the result of the difference between the mean blood glucose levels of the BB X Bb+bb genotypes, the Cohen's calculated effect size d= 0.5, both comparing only the BB x bb group and including the heterozygote together with bb. According to the literature, this value means an effect of medium magnitude. We have added the information to the text of the results in the article.

Cohen J. Statistical power analysis for the behavioral sciences 2d ed. New York: Academic Press. 1988.

4. The authors should elaborate the exclusion criteria which were followed during recruitment of patients in the cohort.

R: The aim of the present study was to analyze the effect of variables that affect the risk of a diabetic individual to develop DR or to present a worse clinical or laboratory condition. The presence of SAH, dyslipidemia, elevation of creatinine were not considered exclusion criteria, but were collected in the clinical evaluation.

Text inserted in the methods: 

R: Inclusion criteria were: diagnosis of DM2 for at least 5 years, being in outpatient care. Exclusion criteria: diagnosis of DM1, insufficient DNA sample or with an inconclusive result in the genotypic analysis. (LINES 117 TO 119)

5. The authors should include the allelic frequencies of SNP rs1544410 by making a query in the 1000 genomes database (https://www.internationalgenome.org), Single Nucleotide Polymorphism database (dbSNP, http://www.ncbi.nlm.nih.gov/SNP) and ClinVar (http://www.ncbi.nlm.nih.gov/clinvar/) database.

R: Detailed information regarding allelic frequency and HWE was included in two paragraphs of the discussion: (LINES 283 TO 299)

The MAF found in this study was 0.47 in the diabetic group and 0.43 in DR patients while in another Brazilian sample from Minas Gerais State (Southeast region) of the country, the MAF found was 0.40 in the group of type 2 diabetics. In the samples of the 1000 genome project, obtained from healthy population, the MAF (A) for the SNP is 0.29. According to the SNP database (ncbi.nlm.gov) in two different Latin American populations the MAF found was 0.36 (n=798) and 0.23 (n=3896), in Europeans it was 0.39 and in African Americans 0.26. Interestingly, in Asian populations the MAF is significantly lower, showing that it is a SNP that is highly influenced by ethnic aspects of the population under analysis. These informations evidenciate similarity between the MAF found in this study and another sample of Brazilian diabetics and a Latin American sample.

The genotypic distribution found deviated from the Hardy Weinberg Equilibrium in both groups, DM2 and DM2 with DR. This finding has not been uncommon in studies involving samples in studies involving the polymorphism in question and diabetes. The HWE imbalance was present in a study conducted in the USA [42] and another in Chile [58]. In a recent meta-analysis that evaluated the effect of rs1544410 on the risk of DM2, of the 37 studies analyzed, 10 showed a deviation from the HWE in samples from different locations around the world [59].

Minor comments

Please recheck the figure legends and statistical tables thoroughly for language errors.

R: As we included new samples in the study, all tables and footnotes were reworked and rewritten. The entire statistical analysis was redone although the main data remained unchanged compared to the original version of the paper.

Overall, results of the present study do not show an association of the rs1544410 polymorphism with the risk of DR causation. However, the authors have found statistically significant association with a biochemical parameter; fasting blood glucose of the DR group. In my opinion, the authors should significantly enrich the results of this manuscript by increasing the sample size and include Cox proportional hazards regression model or logistic regression models to analyse the effect of different risk factors on the causation of DR. In current study, the authors can also include a systematic and detailed meta-analysis to further enrich this manuscript.

R: In fact, our main objective was to investigate whether the studied polymorphism affects metabolic aspects in the clinical context of diabetic retinopathy. Therefore, we analyzed in logistic regression whether the results found in the binary statistics (glycemia and insulin therapy) were in fact influenced by the presence of the b allele. It was not our goal to analyze causes of DR except for the effect of genotypes (which we found no association in the tested scenarios). We believe we met the reviewer's demand as we increased the sample size in both groups and included the logistic regression statistical model as recommended.

The meta-analysis was not included because there are already publications of this nature in the literature on the same subject and covering practically all articles published so far. Our discussion text cites some of them.

---

## [Decision Letter · Decision Letter 1]

17 Jan 2022

Metabolic impact of the VDR rs1544410 in diabetic retinopathy

PONE-D-21-31455R1

Dear Dr. ASSIS,

We’re pleased to inform you that your manuscript has been judged scientifically suitable for publication and will be formally accepted for publication once it meets all outstanding technical requirements.

Kind regards,

Kanhaiya Singh, Ph.D

Academic Editor

PLOS ONE

Additional Editor Comments (optional):

Reviewers' comments:

Reviewer's Responses to Questions

**Comments to the Author**

1. If the authors have adequately addressed your comments raised in a previous round of review and you feel that this manuscript is now acceptable for publication, you may indicate that here to bypass the “Comments to the Author” section, enter your conflict of interest statement in the “Confidential to Editor” section, and submit your "Accept" recommendation.

Reviewer #2: All comments have been addressed

2. Is the manuscript technically sound, and do the data support the conclusions?

Reviewer #2: Yes

3. Has the statistical analysis been performed appropriately and rigorously? 

Reviewer #2: (No Response)

4. Have the authors made all data underlying the findings in their manuscript fully available?

Reviewer #2: Yes

5. Is the manuscript presented in an intelligible fashion and written in standard English?

Reviewer #2: Yes

6. Review Comments to the Author

Reviewer #2: The current submission has examined the metabolic impact of VDR rs1544410 in diabetic retinopathy. The authors have submitted the revised version of the manuscript. The results and the rebuttal letter confirms that all the queries have been addressed.

7. PLOS authors have the option to publish the peer review history of their article (what does this mean?). If published, this will include your full peer review and any attached files.

Reviewer #2: No

---

## [Editor Report · Acceptance letter]

11 Feb 2022

PONE-D-21-31455R1 

Metabolic impact of the VDR rs1544410 in Diabetic Retinopathy

Dear Dr. Assis:

I'm pleased to inform you that your manuscript has been deemed suitable for publication in PLOS ONE. Congratulations! Your manuscript is now with our production department. 

Kind regards, 

on behalf of

Dr. Kanhaiya Singh 

Academic Editor

PLOS ONE